# Associations between Child Snack and Beverage Consumption, Severe Dental Caries, and Malnutrition in Nepal

**DOI:** 10.3390/ijerph17217911

**Published:** 2020-10-28

**Authors:** Neha Zahid, Nehaa Khadka, Madhurima Ganguly, Tanya Varimezova, Bathsheba Turton, Laura Spero, Karen Sokal-Gutierrez

**Affiliations:** 1School of Public Health, University of California, Berkeley, CA 94704, USA; ksokalg@berkeley.edu; 2Department of Epidemiology, Fielding School of Public Health, University of California, Los Angeles, CA 90095, USA; nehaakhadka@gmail.com; 3School of Dentistry, University of California, San Francisco, CA 94143, USA; madhurima.ganguly@gmail.com (M.G.); tvarimezova@gmail.com (T.V.); 4Department of Dentistry, University of Puthisastra, Phnom Penh 12211, Cambodia; bethy.turton@gmail.com; 5Jevaia Foundation, Pokhara 33700, Nepal; Laura@jevaia.org

**Keywords:** oral health, dental caries, malnutrition, Nepal, sugar-sweetened beverages, junk food

## Abstract

The global nutrition transition and increased consumption of sugar-sweetened beverages and ultra-processed snacks have contributed to increasing rates of child obesity and dental caries in developing countries. In Nepal, where child malnutrition rates are high, the relationship between malnutrition and dental caries is poorly understood. This cross-sectional study aims to assess this relationship among a convenience sample of 273 children age six months to less than 12 years in three communities in Nepal, using parent/caregiver interviews, child dental exams, and anthropometric measurements. Fisher’s exact test and independent t-tests examined associations between dietary practices and severe caries and between severe caries and malnutrition, respectively. Children consumed sugar-sweetened beverages and processed snacks frequently: 80% consumed tea with sugar, 60% consumed sweet snacks, and 65% consumed processed savory snacks daily. Overall, 74% of children had untreated tooth decay, and 21% exhibited stunting malnutrition, 14% were underweight, and 6% presented wasting. Significant associations were found between daily consumption of sweets and processed snacks with severe caries and between severe caries and poorer nutritional status. These findings underscore the need to incorporate nutrition and oral health promotion and dental treatment into maternal–child health services and schools and to strengthen policies to reduce children’s access to junk food.

## 1. Introduction

The global “nutrition transition” describes the shift from a traditional, whole-food diet to a highly-processed diet high in sugar, fat, and salt, often referred to as the “Western diet” [1]. Over recent decades, developing countries have experienced large dietary changes associated with increasing income, urbanization, trade liberalization, and marketing [2,3,4]. Nepal has had a gradual shift in diet for the past forty years, but recent decades have seen an accelerated increase in sugar and fat consumption [5,6,7]. Drivers of the dietary changes include child-targeted marketing of processed sweet and savory snack foods and sugar-sweetened beverages (SSBs), sold in and around schools, at very low prices [8,9]. Children are vulnerable to marketing strategies and can develop serious diet-related health conditions including severe dental caries (tooth decay), malnutrition, and obesity, which can have adverse short-term and long-term consequences [4,10].

Dental caries, the most prevalent chronic disease of childhood, is driven by exposure to dietary sugars. Metabolized within the dental biofilm, these sugars create a state of imbalance where net demineralization of the tooth structure occurs. That process, when left unchecked, will lead to cavitation and then chronic infection of the pulp space and surrounding tissues [11,12,13]. Caries can be prevented by limiting bottlefeeding, avoiding feeding young children SSBs and ultra-processed snacks, brushing children’s teeth with fluoride toothpaste starting in infancy, applying topical dental fluoride varnish, and ensuring dental screening and treatment [11,13,14]. The impacts of severe childhood caries can include chronic dental pain, difficulty eating and sleeping, school absence, and poor overall quality of life [10,15,16,17,18]. In Nepal, studies show that upwards of 60% of children experience dental caries [19,20,21,22,23,24]. Nepal also has high rates of child malnutrition—approximately 40% stunting, 30% underweight, and 10% wasting [25]—further increasing children’s risk for infectious disease, limited growth and development, and poorer quality of life [26,27].

Sugar consumption is a common risk factor for dental caries, malnutrition, and obesity [10,24,28]. There have been mixed findings on the association between severe caries and nutritional status. Studies have demonstrated an association between severe caries and malnutrition in some low-income contexts, with caries contributing to mouth pain and chronic inflammation which disrupt normal eating, sleep, and growth [29,30,31,32,33]; associations with obesity in high-income contexts due to shared dietary risk factors [34,35]; and no associations in other studies [36]. There is limited research on these relationships in Nepali populations. Given the current pandemics of child sugar consumption, obesity, undernutrition, and dental caries, there is a need for greater understanding of the relationships among these health risks to guide interventions to improve child health and wellbeing. This study aims to explore the associations between Nepali children’s dietary consumption habits and dental caries and between severe caries and nutritional status.

## 2. Materials and Methods

### 2.1. Study Population and Data Collection

This is a cross-sectional study of dietary practices, nutritional status, and oral health in a convenience sample of 273 Nepali children aged six months to 12 years, conducted in December 2016. Participants were recruited by our local partner organization, Jevaia Foundation, a Nepal-based non-governmental organization working with the Ministry of Health to build a system of community-based dental care in rural Nepal. Participating villages were selected based on the interest, needs, and working capacity of Jevaia Foundation. All families with children from six months to 12 years of age from two rural villages (Puranchaur and Hansapur) and one urban community (Pokhara City) in the Kaski district of Nepal were invited to participate in a community dental health camp. Camp activities included data collection followed by nutrition and oral health education, distribution of toothbrushes and fluoride toothpaste, fluoride varnish application, and referral to dental treatment as needed. This study received ethical approval by the University of California, Berkeley Committee for Protection of Human Subjects (#2010-06-1655), University of California, San Francisco Human Research Protection Program by inter-institutional reliance (#2493-UCSF), and the Nepal Health Research Council (#420/2016).

Trained Jevaia field staff, who were experienced in conducting oral health promotion in Nepal, provided verbal explanation of the study to participating parents/caregivers and children, obtained written informed consent from adults for their own and their children’s participation, and obtained verbal assent from children according to their developmental ability. Field staff interviewed parents/caregivers regarding family demographic characteristics, maternal and child dietary and oral health practices, dental symptoms, and access to dental care (Appendix A). Interview responses were recorded on iPads via the 2015 Qualtrics platform (Provo, UT, USA) and uploaded to a secure server. Trained volunteers measured children’s length/height and weight, in light clothing without shoes, using a stadiometer and professional grade scale (Seca, Chino, CA, USA), according to the World Health Organization (WHO) standards [37], recording to the nearest centimeter and 0.1 kg, respectively. Child dental exams (Appendix A) were performed by two calibrated examiners (inter-examiner kappa scores >0.8) who conducted examinations with children in a supine or sitting position with a headlamp and mirror. Two indices were used to record caries experience, the decayed, missing (due to decay), and filled (due to decay) teeth (dmft+DMFT) index [38] as well as pulpal involvement, ulcer, fistula, and abscess (pufa+PUFA) index [39]. The WHO basic methods for oral health surveys manual was used to determine the threshold for caries diagnosis; a lesion was said to be decayed if dentine was visible by direct vision [38]. Caries experience in the primary and permanent dentitions was recorded separately.

### 2.2. Measures

To assess children’s consumption of snack foods and beverages associated with dental caries, parents were asked about the frequency of their child’s consumption of milk, soda, tea with sugar, sweet snacks, and processed savory snacks. Sweet snacks were defined as sweets, candy, and chocolate, and processed savory snacks were defined as chips and biscuits. For each food or beverage, parents were asked to estimate their child’s frequency of consumption within six categories: Never, Every 2–3 Weeks, Once a Week, 2–3 Times a Week, Daily, and 2–3 Times a Day. Frequency of consumption of milk, soda, tea with sugar, sweets, and processed snacks comprised the predictor variables for the association between dietary consumption and severe caries.

Three indices for severe caries were used as the outcome variables for the association between diet and severe caries and predictor variables for the association between severe caries and malnutrition: (1) Severe Early Childhood Caries (S-ECC), for children with primary dentition (<6 years), defined by any caries in children <3 years, dmft score >4 at age 3, dmft score >5 at age 4, dmft score >6 at age 5, or any dmft in the maxillary anterior incisors [40]. (2) The Significant Caries Index (SiC), for children with permanent dentition (6 years to less than 12 years in this study), defined as the third of the population with the greatest DMFT score [41]. (3) The presence of pulpal involvement, ulceration, fistula or abscess (pufa+PUFA) in primary or permanent dentition (6 months to less than 12 years in this study), an index of severe clinical consequences of untreated deep caries [39]. Using the WHO dental indices for childhood caries, the study population was separated into three subpopulations: younger children (6 months to less than 6 years), older children (6 years to less than 12 years), and the total population (6 months to less than 12 years) [38].

Child nutritional status was determined by the World Health Organization (WHO) growth reference standards for children [37,42]. WHO Anthro (v. 3.2.2) software was used to transform child height and weight data into Height-for-Age Z (HAZ) score, Weight-for-Age Z (WAZ) score, and BMI-for-Age Z (BAZ) score. HAZ and BAZ scores were produced for all ages, while WAZ scores were produced up to age ten. Malnutrition was defined according to WHO as stunting (HAZ ≤ −2), underweight (WAZ ≤ −2), and wasting (BAZ ≤ −2) [37,42].

### 2.3. Data Analysis

Descriptive data were examined using frequencies with mean and standard deviation (SD) for family demographic characteristics, and the responses were stratified by age for child nutrition and oral health characteristics. The responses assessing diet consumption were consolidated into Rarely or Never (combining Never and Every 2–3 Weeks), Weekly (combining Once a Week and 2-3 Times a Week), and Daily (combining Daily and 2–3 Times a Day). The association between frequency of dietary consumption and severe caries (S-ECC, SiC, and pufa+PUFA) was examined by applying Fisher’s exact test, a procedure used for small sample statistics with cells less than a value of 5 [43]. Independent t-tests were used to test the association between severe caries (S-ECC, SiC, and pufa+PUFA) and nutritional status (HAZ, WAZ, and BAZ). Statistical analysis with a value of alpha = 0.05 was the threshold for statistical significance. Data were entered into Microsoft Excel 2016 (Microsoft, Redmond, WA, USA) and exported to SPSS version 24 (IBM, Armonk, NY, USA) and SAS 9.4 (SAS Institute, Cary, NC, USA) for statistical analysis.

## 3. Results

### 3.1. Descriptive Results

Family demographic information is presented in Table 1. The mothers’ average age was approximately 30 years; 22% had education levels less than 5th grade, 50% between 5th and 10th grade, and 28% higher than 10th grade. The average age of children was seven years; 41% percent were younger than six years and 59% were six years or older. Fifty-two percent of children were males, and 48% were females.

Child nutrition and oral health characteristics by age are summarized in Table 2. Among the total child population, 67% consumed milk daily, 3% consumed soda daily, 80% consumed tea with sugar daily, 60% consumed sweet snacks daily, and 65% consumed processed savory snacks daily. While younger children consumed milk more frequently and older children consumed SSBs more frequently, children of all ages consumed sugary tea, sweets, and processed savory snacks with similar frequency. In the overall sample, the mean anthropometric Z scores were −1.14, −0.98, and −0.38 for HAZ, WAZ, and BAZ, respectively, with 21% experiencing stunting malnutrition, 14% underweight, and 6% wasting; older children were more likely to exhibit a lower BAZ score than younger children. The overall frequency of dental caries was 74%, with a greater frequency among older children (86%) compared to younger children (58%). Severe early childhood caries (S-ECC) was seen in 41% of younger children, and severe caries by pufa+PUFA was seen in 20% of younger children and 38% of older children.

Figure 1 summarizes the caries experience by age in the population. The frequency of caries increased steadily with age from one to six years old, reaching 90% frequency, and remaining high thereafter. Similarly, the mean number of decayed teeth (dmft+DMFT) increased from age one to seven years, reaching a mean dmft+DMFT score of nearly 6. The proportion of children with severe early childhood caries (S-ECC) also increased steadily with age, reaching over 50% at age five years. The frequency of pufa+PUFA increased in early childhood beginning at age three, and peaking at age seven with a frequency of 60%.

### 3.2. Association between Dietary Consumption and Severe Caries

Table 3 summarizes the results for associations between dietary consumption and severe caries, by age group. Among younger children, there were statistically significant associations between severe early childhood caries and dietary consumption of milk, SSBs, sugar tea, sweets, and processed snacks. Among older children, there were significant associations observed between the significant caries index and dietary consumption of sweets and processed snacks. Across the entire population, there were associations between pufa+PUFA and dietary consumption of sweets and processed snacks.

### 3.3. Association between Severe Caries and Malnutrition

Table 4 summarizes the results for associations between severe caries and nutrition status, by age group. Across the age groups, there were significant associations between the presence of pufa+PUFA and lower weight-for-age Z score and BMI-for-age Z score, and a trend toward a lower height-for-age Z score. In the older children, there was a significant association between SiC and lower height-for-age Z score. In younger children, S-ECC was not found to be associated with the risk for malnutrition.

## 4. Discussion

This exploratory study examined the diet-related risk factors for severe tooth decay and the association between severe tooth decay and nutritional status in a convenience sample of Nepali children. The traditional diet in the areas of Nepal where the study was conducted is rice with lentils and vegetables (locally known as “daal bhaat”), and traditional snacks include corn or popped corn, roasted soybeans, beaten rice, and foods made from locally grown grains such as wheat and millet [44]. The majority of children surveyed also consumed sweets, processed snacks, sugar-sweetened tea, and other SSBs on a daily basis, in both early childhood (under age 6) and middle childhood (age 6–12). This study provides additional evidence of the nutrition transition in both urban and rural areas of Nepal, and that unhealthy snack foods and beverages have become a daily staple of the diet for many children [5,6].

For children under six years with primary dentition, severe early childhood caries was associated with frequent milk consumption as well as frequent consumption of sugary drinks (SSBs and sugar tea) and snacks (sweet and processed). It appears that younger children who never consumed milk experienced a greater frequency of severe caries than those who consumed milk daily or weekly. At the same time, children who consumed sugary beverages and junk food on a daily or weekly basis had a greater frequency of severe caries than those who never consumed these products. For older children (who had mixed dentition), severe caries was associated only with frequent consumption of sweet and processed snacks. These findings suggest that frequent consumption of sweet and processed snacks may be hazardous to the oral health of children of all ages and that sugar-sweetened beverages as well as non-nutritious snacks may be particularly hazardous to children’s oral health in early childhood. This finding aligns with existing evidence that primary teeth are particularly susceptible to rapid progression of caries due to their smaller size and bulbous shape which creates a larger surface area to volume ratio, thinner enamel, and generally poorer oral hygiene in early childhood [45,46]. Additionally, our results indicate frequent milk consumption in early childhood may serve as a protective factor for developing severe early childhood caries, though the direction of this association could not be determined. Generally, milk consumption has been shown to be protective against ECC as the calcium aids in tooth mineralization [47,48], and children who drink more milk tend to drink lower quantities of sugar-sweetened beverages that cause caries [49,50]. However, some high-frequency of milk consumption has been found to be cariogenic—particularly bottlefeeding in bed, bottlefeeding over two years of age, and drinking sugar-sweetened milk [15,51].

In this study population, caries was observed to begin in the first two years of life and increased steadily in frequency and severity over early childhood. Tooth eruption began as early as six months of age, with the first sign of tooth decay at 12 months of age. At age seven, the mean dmft+DMFT (5.9) and the frequency of pufa+PUFA (58%) were at their highest. The decline in mean dmft+DMFT and pufa+PUFA frequency starting after age seven is likely due to the normal exfoliation of primary teeth that were decayed. From infancy to age 12, the majority of children (74%) had tooth decay, but it was the approximately 1/3 of children with severe caries who were at significantly increased risk for the adverse impact of caries on their nutritional status.

While the frequency of malnutrition in this sample was lower than that in other studies [25], the rates are still alarming. In the total sample, 20% of children were stunted, 14% were underweight, and 6% were wasted, indicating the presence of both chronic and acute malnutrition. Across both age groups, severe caries by pufa+PUFA—an indication of deep dental infection, cavitation, and likelihood of dental pain—was associated with lower WAZ and BAZ, suggesting a risk for acute malnutrition, and in older children, severe caries by SiC was associated with lower HAZ, suggesting a risk for chronic malnutrition. This study adds to the growing literature from developing countries, demonstrating that severe dental caries may exacerbate the risk for childhood malnutrition, especially in early childhood [29,32,33,36]. The pathophysiologic mechanisms by which severe caries may contribute to malnutrition include chronic infection/inflammation and dental pain suppressing children’s appetite and growth, loss of tooth structure inhibiting children’s ability to chew nutritious foods, and increasing use of non-nutritious sweets to pacify children’s chronic pain [32]. The association between severe caries and malnutrition underscores the importance of caries prevention and early intervention, which may not only improve children’s oral health but may also improve their nutritional status, growth, and development.

While the maternal-child health field and child malnutrition literature has continued to focus on poverty, insufficient macro- and micro-nutrients, and infectious diseases as the main contributors to child malnutrition, the possible contributions of excessive sugar consumption and dental caries have not been acknowledged [52]. There needs to be greater recognition of the modern threats to childhood nutritional status from frequent consumption of junk food and SSBs as well as severe early childhood caries. Maternal–child health interventions to promote child nutrition should include a focus on promoting breastfeeding and milk consumption, preventing children’s daily consumption of junk food and SSBs, promoting children’s oral health, and screening and treating children for dental caries, particularly in early childhood [53,54]. Nutrition and oral health interventions can be delivered through maternal-child health clinics and early childhood education programs [45], and policy interventions could tax sugary snacks and beverages, prohibit the sale of unhealthy snack foods and drinks in and around health facilities and schools, and promote universal access to clean water and dental care [46,55,56].

This study has limitations including the use of a cross-sectional analysis to identify associations but cannot demonstration causation. The use of a convenience sample may not be generalizable to the entire population, and a small sample size limits the power of the statistical analysis and the ability to conduct multivariable analysis that can control for confounding. Analysis of severe caries is also limited by the aggregation of dmft and DMFT in children with mixed dentition. Furthermore, this study assessed dental caries as a primary outcome and exposure and did not include a complete oral hygiene evaluation. Oral hygiene was evaluated by self-report yet we did not assess toothpaste usage, plaque scores, or gingival inflammation. Moreover, the use of broad age ranges for analyzing the association between dietary consumption and severe caries and malnutrition limits the ability to make conclusions about these relationships in narrower age strata. In addition, due to the limited time available for interviews, the questionnaire addressed a small number of caries-related snack foods and beverages, rather than a comprehensive dietary assessment, which could have provided greater dietary detail. Finally, dietary behaviors as reported by parents/caregivers may have underestimated children’s consumption of junk food due to acceptability bias or lack of awareness of their children’s consumption outside of parental supervision. Despite these limitations, this exploratory study is useful for providing a foundation to inform further hypothesis formation about caries and malnutrition in Nepal. Future research should use more robust sampling and statistical methodology to better understand the directionality of sugar consumption, caries, and nutrition-related disease among Nepali children.

## 5. Conclusions

This cross-sectional study of the nutrition and oral health of Nepali children aged six months to less than 12 years demonstrated a high prevalence of the daily consumption of junk food and sugar-sweetened beverages, significant associations between the frequency of junk food consumption and severe dental caries, and significant associations between severe dental caries and the risk for malnutrition. Maternal–child health programs, early childhood education, and schools should incorporate combined nutrition and oral health interventions to promote the traditional diet, oral hygiene, and dental screening and treatment, as well as advocacy for policies to limit the marketing of unhealthy products to children to ensure that children’s environments promote good nutrition and oral health.

## Figures and Tables

**Figure 1 ijerph-17-07911-f001:**
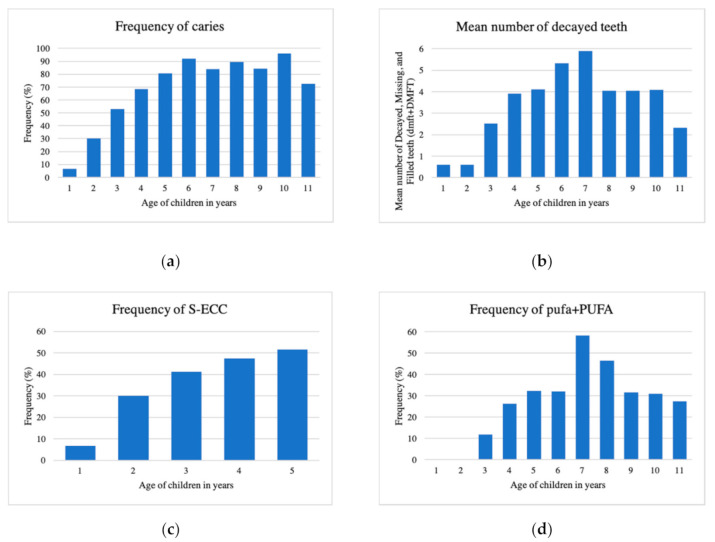
Caries experience by age. A combination of four graphs showing: (**a**) Frequency of caries; (**b**) Mean number of decayed teeth, defined as decayed, missing, or filled teeth (dmft+DMFT); (**c**) Frequency of severe early childhood caries (S-ECC, i.e., caries in children under age 6); (**d**) Frequency of pulpal involvement, ulceration, fistula, or abscess (pufa+PUFA) by age.

**Table 1 ijerph-17-07911-t001:** Family Demographic Characteristics.

Category	Mean ± SD or No. (%)
MOTHERS’ DEMOGRAPHICS (N = 209)	
Age	30.6 ± 6.4
Education level	
Less than 5th grade	47 (22.3)
Between 5th and 10th grade	103 (49.5)
Greater than 10th grade	60 (28.2)
CHILDREN’S DEMOGRAPHICS (N = 273)	
Age	7.0 ± 3.0
6 months to less than 6 years	111 (40.7)
6 years to less than 12 years	162 (59.3)
Sex	
Male	141 (52.0)
Female	132 (48.0)

**Table 2 ijerph-17-07911-t002:** Dietary Frequency, Nutrition, and Oral Health Characteristics by Age.

Category	No. (%) or Mean ± SD			P
	Younger Children (6 months to less than 6 years)	Older Children (6 years to less than 12 years)	Total Population (6 months to less than 12 years)	
	N = 111	N = 162	N = 273	
DIETARY FREQUENCY CHARACTERISTICS				
Milk Consumption				0.045
Rarely or Never	10 (9)	22 (14)	32 (12)	
Weekly	16 (14)	39 (24)	55 (20)	
Daily	84 (76)	99 (61)	183 (67)	
SSBs Consumption ^a^				<0.001
Rarely or Never	27 (24)	9 (6)	36 (13)	
Weekly	78 (70)	147 (91)	225 (83)	
Daily	5 (5)	3 (2)	8 (3)	
Sugar Tea Consumption				0.079
Rarely or Never	17 (15)	12 (7)	29 (11)	
Weekly	12 (11)	12 (7)	24 (9)	
Daily	81 (73)	137 (85)	218 (80)	
Sweets Consumption				0.104
Rarely or Never	10 (9)	5 (3)	15 (6)	
Weekly	33 (30)	53 (33)	86 (32)	
Daily	66 (60)	98 (61)	164 (60)	
Processed Snacks Consumption				0.078
Rarely or Never	6 (5)	2 (1)	8 (3)	
Weekly	38 (34)	49 (30)	87 (32)	
Daily	66 (60)	111 (69)	177 (65)	
NUTRITION CHARACTERISTICS				
Height-for-age Z score (HAZ)	−1.28 ± 1.33	−1.04 ± 1.09	−1.14 ± 1.20	0.099
HAZ ≤ −2	28 (25)	30 (19)	58 (21)	
Weight-for-age Z Score (WAZ) ^b^	−0.91 ± 1.11	−1.06 ± 1.19	−0.98 ± 1.15	0.348
WAZ ≤ −2 ^b^	17 (15)	20 (12)	37 (14)	
BMI-for-age Z score (BAZ)	−0.12 ± 0.96	−0.55 ± 1.11	−0.38 ± 1.07	0.001
BAZ ≤ −2	3 (3)	12 (8)	15 (6)	
ORAL HEALTH CHARACTERISTICS				
Frequency of caries	64 (58)	139 (86)	203 (74)	
dmft+DMFT ^c^ in total population	3.0 ± 3.9	4.25 ± 3.76	3.75 ± 3.84	0.010
dmft+DMFT ^c^ in population with decay	5.2 ± 3.8	4.96 ± 3.60	4.99 ± 3.58	
Frequency of severe caries ^d^	45 (41)	54 (33)	99 (36)	
Frequency of pufa+PUFA ^e^	22 (20)	62 (38)	84 (31)	

^a^ Excludes tea with sugar; ^b^ Weight-for-age Z (WAZ) score is depicted for ages 6 months to less than 10 years only; ^c^ Decayed, missing, or filled teeth (dmft+DMFT); ^d^ Severe caries is characterized by severe early childhood caries (S-ECC) for younger children and by the significant caries index (SiC) for older children; ^e^ Pulpal involvement, ulceration, fistula, or abscess in permanent or primary dentition (pufa+PUFA).

**Table 3 ijerph-17-07911-t003:** Association between Dietary Consumption and Severe Caries.

Dietary Consumption	N	Frequency (%)	P d	N	Frequency (%)	P d	N	Frequency (%)	P d
	S-ECC ^a^ (6 months to less than 6 years)			SiC ^b^ (6 years to less than 12 years)			pufa+PUFA ^c^ (6 months to less than 12 years)			
Milk			0.002			0.428			0.111
Never	15	11 (73)		32	8 (25)		47	14 (30)	
Weekly	59	16 (27)		96	33 (34)		155	42 (27)	
Daily	36	18 (50)		32	13 (41)		68	28 (41)	
SSBs ^e^			<0.001			0.263			0.266
Never	78	22 (28)		77	21 (27)		155	41 (26)	
Weekly	29	21 (72)		80	30 (38)		109	38 (35)	
Daily	3	1 (33)		2	1 (50)		5	2 (40)	
Sugar Tea			0.004			0.939			0.216
Never	25	5 (20)		19	7 (37)		44	9 (20)	
Weekly	47	27 (57)		99	32 (32)		146	50 (34)	
Daily	38	12 (32)		43	14 (33)		81	24 (30)	
Sweets			<0.001			0.011			0.009
Never	21	1 (5)		21	5 (24)		42	6 (14)	
Weekly	39	14 (36)		67	16 (24)		106	29 (27)	
Daily	49	29 (60)		68	32 (47)		117	45 (38)	
Processed Snacks			0.033			0.002			0.008
Never	19	4 (21)		13	2 (15)		32	4 (13)	
Weekly	37	12 (32)		60	12 (20)		97	25 (26)	
Daily	54	28 (52)		89	40 (45)		143	54 (38)	

^a^ Severe early childhood caries (S-ECC); ^b^ Significant Caries Index (SiC); ^c^ Pulpal involvement, ulceration, fistula or abscess (pufa+PUFA); ^d^ Fisher’s Exact Test (2-tailed); ^e^ Excludes tea with sugar.

**Table 4 ijerph-17-07911-t004:** Association between Severe Caries and Nutritional Status.

Age Group	Severe Caries	Malnutrition Index	N	Mean	Mean Difference from Reference; (95% CI)	P
Younger Children (6 months to less than 6 years)	S-ECC ^a^					
		Height-for-age (HAZ)	111			
	No			−1.33	Ref	
	Yes			−1.20	0.13; (−0.64, 0.38)	0.609
		Weight-for-Age (WAZ)	111			
	No			−0.89	Ref	
	Yes			−0.93	−0.05; (−0.39, 0.48)	0.835
		BMI-for-age (BAZ)	111			
	No			−0.05	Ref	
	Yes			−0.23	−0.18; (−0.19, 0.55)	0.333
Older Children (6 years to less than 12 years)	SiC ^b^					
		Height-for-age (HAZ)	162			
	No			−0.89	Ref	
	Yes			−1.36	−0.47; (0.12, 0.83)	0.009
		Weight-for-age (WAZ) ^c^	104			
	No			−0.97	Ref	
	Yes			−1.19	−0.22; (−0.25, 0.69)	0.346
		BMI-for-age (BAZ)	162			
	No			−0.58	Ref	
	Yes			−0.51	0.07; (−0.44, 0.30)	0.709
Total Population (6 months to less than 12 years)	pufa+PUFA ^d^					
		Height-for-age (HAZ)	273			
	No			−1.05	Ref	
	Yes			−1.34	−0.28; (−0.02, 0.59)	0.070
		Weight-for-age (WAZ)	215			
	No			−0.87	Ref	
	Yes			−1.24	−0.37; (0.04, 0.70)	0.028
		BMI-for-age (BAZ)	273			
	No			−0.28	Ref	
	Yes			−0.60	−0.32 (0.04, 0.59)	0.025

^a^ Severe early childhood caries (S-ECC); ^b^ Significant Caries Index (SiC); ^c^ WAZ index used for ages 6 years to less than 10 years; ^d^ Pulpal involvement, ulceration, fistula, or abscess (pufa+PUFA).

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
