# Peer review of "Associations between Child Snack and Beverage Consumption, Severe Dental Caries, and Malnutrition in Nepal"

_ijerph, 2020, doi:10.3390/ijerph17217911_

Round 1
Reviewer 1 Report
This is an interesting study, but the major limitation is a lack of oral hygiene evaluation. This limitation should be mentioned in the discussion.
I also think that categories of mother's education should be described or defined in more detail.
What was the age of children ? Were 12-year-old children included as mentioned in the materials and methods section, or only children up to the age of 11 participated ("less than 12"?).
I am not sure if this is a proper conclusion:
"For children under 6 years with primary dentition, severe early childhood caries was negatively associated with frequent milk consumption and positively associated with frequent consumption of sugary drinks (SSBs and sugar tea) and snacks (sweet and processed)." Fisher's test for 2/3 contingency tables was used and p was significant. But do we know what accounted for this result? Weekly consumption of milk seems more beneficial that daily consumption when I look on frequencies. Weekly consumption of sugary drinks seems more harmful than daily consumption.
What does the column 5 in table 3 contain? What does the column 4 in table 4 contain? Please add some explanations.
Author Response
Dear Dr. Sabbah and Reviewer 1,
Thank you for the recent review of our paper, “Associations Between Child Snack and Beverage Consumption, Severe Dental Caries and Malnutrition in Nepal.” We appreciate the detailed comments from the Reviewers, and welcome the opportunity to revise and resubmit the manuscript. We are submitting a revised manuscript with responses to each of the reviewer comments. To guide you to the changes in the manuscript, we have highlighted in yellow in the manuscript each
change made in response to the comments; and we include a summary below with the comments from the Reviewers followed by our brief response in italics with the line number(s) on which the edits appear.
Reviewer 1:
1. This is an interesting study, but the major limitation is a lack of oral hygiene evaluation. This limitation should be mentioned in the discussion.
Response: On Lines 270-273 we have updated the limitations section of the discussion to address the lack of oral hygiene evaluation: “Furthermore, this study assessed dental caries as a primary outcome and exposure and did not
include a complete oral hygiene evaluation. Oral hygiene was evaluated by self-report yet we did not assess toothpaste usage, plaque scores, or gingival inflammation.”
2. I also think that the categories of mother’s education should be described or defined in more detail.
Response: The three education levels used (less than 5th standard; between 5th and 10th standard; and greater than 10th standard) are typical in the Nepali education system. Less than 5th standard corresponds to primary education;
between 5th and 10th corresponds to up to high school education; and greater than 10th standard corresponds to high school education or higher. “Standard” is interchangeable with “grade” so in our text under 3.1 Descriptive Results and our Table 1, we have replaced “standard” with “grade.” (Lines 143, 144).
3. What was the age of children ? Were 12-year-old children included as mentioned in the materials and methods section, or only children up to the age of 11 participated (“less than 12”?”.
Response: The age of the children was 6 months to less than 12 years, or up to the age of 12 (12 year old children were not included). This has been clarified in the methods section on Lines 114 and 116-117 by changing: “(6-12 years in this study)” to “(6 years to less than 12 years in this study)” and “(0-12 years in this study)” to “(6 months to less than 12 years in this study)” We have also corrected this age range in the abstract (Line 19): These changes are consistent with the language used to describe our study population on Lines 119-120 and in Tables
1-4.
4. I am not sure if this is a proper conclusion:
"For children under 6 years with primary dentition, severe early childhood caries was negatively associated with frequent milk consumption and positively associated with frequent consumption of sugary drinks (SSBs and sugar tea) and snacks (sweet and processed)." Fisher's test for 2/3 contingency tables was used and p was significant. But do we know what accounted for this result? Weekly consumption of milk seems more beneficial that daily consumption when I look on frequencies. Weekly consumption of sugary drinks seems more harmful than daily consumption.
Response: Thank you for your comment. You are correct in that Fisher’s test does not quite tell us the direction of the association and so to avoid confusion, we have removed the terms “positive” and “negative” and instead provide an extended interpretation of the results. The following changes with their respective sections have been made:
• Results: “Among younger children, there were statistically significant associations observed between severe early childhood caries and dietary consumption of milk, SSBs, sugar tea, sweets and processed snacks. Among older
children, there were significant associations observed between significant caries index and dietary consumption of sweets and processed snacks.” (Lines 179-182).
• Discussion: “For children under 6 years with primary dentition, severe early childhood caries was associated with frequent milk consumption as well as frequent consumption of sugary drinks (SSBs and sugar tea) and snacks
(sweet and processed). It appears that younger children who never consumed milk experienced a greater frequency of severe caries than those who consumed milk daily or weekly. At the same time, children who consumed sugary beverages and junk food on a daily or weekly basis had a greater frequency of severe caries than those who never consumed these products.” (Lines 209-214)
Additionally, in this study of a convenience sample, we used a Fisher’s exact test to explore the possible associations between severe caries and dietary consumption and are thus unable to determine the direction of association. Future
studies should use more robust sampling and statistical methodology to determine the direction of association and the effect of weekly and daily consumption habits on developing caries. We have included the following in the discussion section to call for more research in this area:
• “Despite these limitations, this exploratory study is useful for providing a foundation to inform further hypothesis formation about caries and malnutrition in Nepal. Future research should use more robust sampling and statistical
methodology to better understand the directionality of sugar consumption, caries and nutrition-related disease among Nepali children.” (Lines 280-284).
5. What does the column 5 in table 3 contain? What does the column 4 in table 4 contain? Please add some explanations. Response: It appears that the table cell sizes in our tables were reformatted by the IJERPH after our submission and the values became unclear. Thank you for pointing this out. We have readjusted the cell sizes so that the values are now clear. Column 5 in Table 3 represents the N or count for the children in the 6 years to less than 12 years age group for each dietary consumption variable. Column 4 in Table 4 represents the N or count for each malnutrition index in each child group (younger children, older children, and total population).
Thank you again very much for your review and reconsideration of our revised manuscript. We look forward to hearing your decision.
Reviewer 2 Report
This study explored the association between snacking, caries and malnutrition in a convenience sample of Nepalese Children
Abstract
- The aim of the study was not clear from the abstract
Introduction
- It is obvious that dental caries share common risk factors high sugar diet, thus easy to hypothesise caries to be associated with obesity. As this study intends to look at the association between malnutrition and caries, it is important to provide theoretical/conceptual framework for this hypothesised association
- Aim indicates that the study explores association between dietary consumption and oral health outcomes while 'dental caries' was the only outcome assessed in this study
Methods
- This reviewer isn't sure if the data is still valid as the study data is approximately 4 years old.
- More information required on how many examiners were involved and what was the threshold/method of caries diagnosis
- WHO growth reference standard might not be applicable to Nepalese children. Nepalese national standard would be more appropriate
- The measure/instrument used in the paper could be provided as a supplementary file
- Seems like a typo, a p value =0.05 used as a threshold for significance?
- It is very important to address the effect of confounding variables when exploring association between the diet and dental caries as diet is just one factor that is required for dental caries causation. Multivariable analysis by including all those variables (SES, oral hygiene habits etc) need to be conducted to address confounding problem
- It is not clear why the data from excel was imported both into SAS and SPSS. Which package was ued for analysis?
Results
- It is not logical to claim nutrition (table 2)was assessed in the study when just few closed ended questions were used to assess dietary intake of cariogenic foods
- Caries in Table 2 is presented as dmft/DMFT. Should it be dmft+DMFT as caries in both the dentitions was assessed
- Table 2 is not related to the aim of the study, its relevance is not clear
- Figures are not clear. Figure c is incomplete
- Table 3 could be improved. Without providing n (%) for subjects without S-ECC, the interpretation related to the association between the dietary variables and severe caries is incomplete.
Discussion
- Discussion is superficial and doesn't discuss all findings comprehensively. For instance, the contrasting direction of associations of milk consumption and SSB consumption with caries needs further discussion. Both frequency milk and SSB consumption could be detrimental
Author Response
Dear Dr. Sabbah and Reviewer 2,
Thank you for the recent review of our paper, “Associations Between Child Snack and Beverage Consumption, SevereDental Caries and Malnutrition in Nepal.” We appreciate the detailed comments from the Reviewers, and welcome the opportunity to revise and resubmit the manuscript. We are submitting a revised manuscript with responses to each of thereviewer comments. To guide you to the changes in the manuscript, we have highlighted in yellow in the manuscript each
change made in response to the comments; and we include a summary below with the comments from the Reviewersfollowed by our brief response in italics with the line number(s) on which the edits appear.
Reviewer 2:
1. This study explored the association between snacking, caries and malnutrition in a convenience sample of Nepalese
Children.Response: Thank you for your clear summary of the study.
2. Abstract: The aim of the study was not clear from the abstract.
Response: To clarify the aim of the study, we have revised the abstract to read: “In Nepal, where child malnutrition rates are high, the relationship between malnutrition and dental caries is poorly understood. This cross-sectional study
aims to assess this relationship among a convenience sample of 273 children age 6 months to 12 years in 3 communities in Nepal, using parent/caregiver interviews, child dental exams and anthropometric measurements.”
(Lines 17-21)
3. Introduction: It is obvious that dental caries share common risk factors high sugar diet, thus easy to hypothesise caries to be associated with obesity. As this study intends to look at the association between malnutrition and caries, it is
important to provide theoretical/conceptual framework for this hypothesised association.
Response: We have revised the following on Lines 59-61 to explain the conceptual framework for the possible association between caries and malnutrition. It now reads: “Studies have demonstrated an association between severe caries and malnutrition in some low-income contexts, with caries contributing to chronic inflammation and mouth pain which disrupt normal eating, sleeping and growth [29-33]; associations with obesity in high-income contexts due to shared dietary risk factors [34, 35]; and no associations in other studies [36].
4. Introduction: Aim indicates that the study explores association between dietary consumption and oral health outcomes while “dental caries” was the only outcome assessed in this study.
Response: To correct for this, we have changed oral health outcomes with dental caries in the introduction section:
“This study aims to explore the associations between Nepali children’s dietary consumption habits and dental caries and between severe caries and nutrition status.” (Line 66).
5. Methods: This reviewer isn’t sure if the data is still valid as the study data is approximately 4 years old.
Response: While we acknowledge that our data was collected in 2016, we want to emphasize that this is an exploratory study looking at associations between dietary habits, caries, and malnutrition at one point in time. We avoid making generalizations about the entirety of Nepal and instead, this study explores these associations with the purpose of informing future study design. Given the relative lack of data rendered from this particular setting, rural Nepal, the findings are of note and interest. Additionally, we believe strongly that data collected in Nepal in December 2016 is still valid for publication 4 years later in 2020. Health measures in Nepal change at a very slow rate. In the past 10-20 years that our team has been working in Nepal, we’ve observed that oral health measures have remained similar over time. The children’s oral health data collected for this study in 2016 is remarkably similar to data that we collected in 2 other regions of Nepal in 2010; and the study data collected in 2010 was determined by IJERPH to be valid and it was published by IJERPH 9 years after the data was collected, in 2019 (Tsang et al, 2019).
6. Methods: More information required on how many examiners were involved and what was the threshold/method of caries diagnosis
Response: We clarified that two examiners performed the child dental exams and we added the following sentence with a reference to explain our threshold for caries diagnosis: “The WHO basic methods for oral health surveys manual was used to determine the threshold for caries diagnosis; a lesion was said to be decayed if dentine was visible by direct vision [38].” (Lines 97-99).
Methods: WHO growth reference standard might not be applicable to Nepalese children. Nepalese national standard would be more appropriate
Response: Thank you for your comment. The WHO growth standards have been developed intentionally to apply to all populations so that individual countries do not need to develop their own standards and that nutritional status can be
compared across countries. According to the 2006 WHO child growth standards and the identification of severe acute malnutrition in infants and children, “The new WHO growth standards confirm earlier observations that the effect of
ethnic differences on the growth of infants and young children in populations is small compared with the effects of the environment. Studies have shown that there may be some ethnic differences among individuals, but for practical
purposes they are not considered large enough to invalidate the general use of the WHO growth standards population as a standard in all populations.” Thus, it is acceptable to use the WHO growth standard to assess the nutritional
status of our sample of Nepali children. (See Refs 37 and 42).
7. Methods: The measure/instrument used in the paper could be provided as a supplementary file
Response: We have included the mother and child interview forms and the odontogram used for dental as supplementary files:
File S1: Mother Interview Form
File S2: Child Interview Form
File S3: Odontogram for dental examination
8. Methods: Seems like a typo, a p value =0.05 used as a threshold for significance?
Response: There is a subtle but important difference between alpha and the p-value. Alpha is the significance level: it is the probability that one will make the mistake of rejecting the null hypothesis when in fact it is true. The p-value
measures the probability of getting a more extreme value than the one produced from the experiment/analysis. If the 3 p-value is greater than alpha, one can accept the null hypothesis. So, it is statistically correct to use alpha in this
case.
9. Methods: It is very important to address the effect of confounding variables when exploring association between the diet and dental caries as diet is just one factor that is required for dental caries causation. Multivariable analysis by
including all those variables (SES, oral hygiene habits etc) need to be conducted to address confounding problem Response: Thank you for your comment and acknowledging the importance of addressing confounding. Given the lack
of data regarding caries and malnutrition in Nepal, the purpose of this study among a convenience sample of Nepali children was to explore the associations between dietary consumption and caries and caries and malnutrition.
Because our sample is not random, using a multivariable model will introduce bias and may not add to our understanding of the situation. The most appropriate application of our findings is instead to inform a future causal
model that can be assessed with more robust sampling and statistics.
Additionally, our small sample size limited our ability to control for confounding in a multivariable model. Instead, we stratified by age as a way to control for confounding variables and we used Fisher’s exact test for the association
between dietary consumption and severe caries, a test specific for small sample sizes. We have revised our limitations section to specifically acknowledge our limited ability to control for confounding:
“The use of a convenience sample may not be generalizable to the entire population and a small sample size limits the power of the statistical analysis and the ability to conduct multivariable analysis that can control for confounding.”
(Lines 268-269).
Also, we included in our discussion section that next steps in this field should use this study as a foundation for future hypothesis formation:
“Despite these limitations, this exploratory study is useful for providing a foundation to inform further hypothesis formation about caries and malnutrition in Nepal. Future research should use more robust sampling and statistical
methodology to better understand the directionality of sugar consumption, caries and nutrition-related disease among Nepali children.” (Lines 280-284)
10. Methods: It is not clear why the data from excel was imported both into SAS and SPSS. Which package was used for analysis?
Response: Two co-authors together conducted different analyses using the software they were most proficient at. NZ conducted the descriptive analysis (Tables 1 and 2) and the analysis between severe caries and malnutrition (Table 4)
in SPSS and NK conducted the analysis between dietary consumption and severe caries (Table 3) in SAS. Both used the same dataset imported from Excel.
11. Results: It is not logical to claim nutrition (table 2)was assessed in the study when just few closed ended questions were used to assess dietary intake of cariogenic foods
Response: We have changed the title of Table 2 to read, “Dietary Frequency, Nutrition, and Oral Health Characteristics by Age” to make this more specific to the results of our study.
12. Results: Caries in Table 2 is presented as dmft/DMFT. Should it be dmft+DMFT as caries in both the dentitions was assessed
Response: We have changed “dmft/DMFT” to “dmft+DMFT” and “pufa/PUFA” to “pufa+PUFA” in all instances in the text, in Tables 2-4, and in Figure 1 (See highlighted text).
13. Results: Table 2 is not related to the aim of the study, its relevance is not clear
Response: Table 2 provides descriptive data on the dietary frequency, nutrition/anthropometric statuses, and caries and oral health characteristics for the three age groupings assessed in this study. We believe this descriptive table is
essential to provide the reader an overview of nutrition and oral health characteristics of this study population before 4 looking at the results of our two primary analyses (Table 3 and Table 4). Descriptive results are an important element of analysis as supported by literature on epidemiological investigations (Rothman et al, 2008).
Results: Figures are not clear. Figure c is incomplete
Response: It appears that the figure sizes were reformatted by IJERPH after our submission. They have now been adjusted so that all four figures can be clearly seen. Figure C depicts the frequency of severe early childhood caries
which is only relevant for children ages 6 months to less than 6 years in this study. This is why the age stops at 5, where 5 represents children ages 5 to 6.
14. Results: Table 3 could be improved. Without providing n (%) for subjects without S-ECC, the interpretation to the association between the dietary variables and severe caries is incomplete.
Response: We believe there may be a misunderstanding in the interpretation of our Table 3. The results summarize the association between dietary consumption and severe caries by age group using Fisher’s exact test, which compares the frequencies across different groups (similar to Chi-square). For example, we observe a statistically significant difference between the frequency of severe early childhood caries among younger children who never consume milk, weekly consume milk, and daily consume milk. The Fisher’s exact test does not provide information on the direction of this association but based on the frequencies, it appears that children who never consume milk experience a greater frequency of severe early childhood caries than children who consume milk daily or weekly. To clarify this interpretation in the text, we have removed the terms “positive” and “negative” and instead provide an extended interpretation of the results. The following changes with their respective sections have been made:
Results: “Among younger children, there were statistically significant associations observed between severe early childhood caries and dietary consumption of milk, SSBs, sugar tea, sweets and processed snacks. Among older children, there were significant associations observed between significant caries index and dietary consumption of sweets and processed snacks.” (Lines 179-182).
Discussion: “For children under 6 years with primary dentition, severe early childhood caries was associated with frequent milk consumption as well as frequent consumption of sugary drinks (SSBs and sugar tea) and snacks (sweet
and processed). It appears that younger children who never consumed milk experienced a greater frequency of severe caries than those who consumed milk daily or weekly. At the same time, children who consumed sugary beverages
and junk food on a daily or weekly basis had a greater frequency of severe caries than those who never consumed these products.” (Lines 209-214)
15. Discussion: Discussion is superficial and doesn’t discuss all findings comprehensively. For instance, the contrasting direction of associations of milk consumption and SSB consumption with caries needs further discussion. Both
frequency milk and SSB consumption could be detrimental
Response: We have added the following to the discussion section (Lines 222-229) to comprehensively discuss our finding of milk consumption and caries. We have also included 5 more references to discuss the existing literature
(See Refs 47-51):
“Additionally, our results indicate frequent milk consumption in early childhood may serve as a protective factor for developing severe early childhood caries, though the direction of this association could not be determined. Generally,
milk consumption has been shown to be protective against ECC as the calcium aids in tooth mineralization [47,48], and children who drink more milk tend to drink lower quantities of sugar-sweetened beverages that cause caries [49,50]. However, some high-frequency of milk consumption has been found to be cariogenic - particularly bottlefeeding in bed, bottlefeeding over 2 years of age, and drinking sugar-sweetened milk [15,51].”
Thank you again very much for your review and reconsideration of our revised manuscript. We look forward to hearing your decision.
Reviewer 3 Report
- This study aims to explore the associations between dietary consumption habits and oral health outcomes and between severe caries and nutrition status in Nepali children.
- Nutrition, oral health and caries are multiple causations. Personal demographic characteristic of children and their parents, dietary patterns, oral health behaviors and practices and use of dental services are possible associated to these healthy issues.
- It is necessary that to considering the multi-factors effect by statistical modelling. However, in this paper, the authors did not apply any multivariate statistical methods in analysis. Is it possible that the authors considered to add some modelling methods to deal with this paper, and it would make the result more reality.
- The age distribution of subject in this study is quiet large. The dental status of preschool and school children are total different, which is temporary and permanent teeth separately. And the possible causations related to the temporary and permanent teeth could be different, so the authors should consider two models separately.
- The sample size of this article is so limited. it would be more suitable if the authors can increase sample size.
Author Response
Dear Dr. Sabbah and Reviewer 3,
Thank you for the recent review of our paper, “Associations Between Child Snack and Beverage Consumption, Severe Dental Caries and Malnutrition in Nepal.” We appreciate the detailed comments from the Reviewers, and welcome the opportunity to revise and resubmit the manuscript. We are submitting a revised manuscript with responses to each of the reviewer comments. To guide you to the changes in the manuscript, we have highlighted in yellow in the manuscript each
change made in response to the comments; and we include a summary below with the comments from the Reviewers followed by our brief response in italics with the line number(s) on which the edits appear.
Reviewer 3:
1. This study aims to explore the associations between dietary consumption habits and oral health outcomes and between severe caries and nutrition status in Nepali children.
Response: Thank you for your summary of the study aims.
2. Nutrition, oral health and caries are multiple causations. Personal demographic characteristic of children and their parents, dietary patterns, oral health behaviors and practices and use of dental services are possible associated to these healthy issues.
Response: Thank you for your summary of the results.
3. It is necessary that to considering the multi-factors effect by statistical modelling. However, in this paper, the authors did not apply any multivariate statistical methods in analysis. Is it possible that the authors considered to add some modelling methods to deal with this paper, and it would make the result more reality.
Response: Thank you for your comment and acknowledging the importance of addressing confounding. Given the lack of data regarding caries and malnutrition in Nepal, the purpose of this study among a convenience sample of Nepali children was to explore the associations between dietary consumption and caries and aries and malnutrition. Because our sample is not random, using a multivariable model will introduce bias and may not add to our understanding of the situation. The most appropriate application of our findings is instead to inform a future causal
model that can be assessed with more robust sampling and statistics. Additionally, our small sample size limited our ability to control for confounding in a multivariable model. Instead, we stratified by age as a way to control for
confounding variables. We have revised our limitations section to specifically acknowledge our limited ability to control for confounding:
“The use of a convenience sample may not be generalizable to the entire population and a small sample size limits the
power of the statistical analysis and the ability to conduct multivariable analysis that can control for confounding.”(Lines 268-269)
Also, we included in our discussion section that next steps in this field should use this study as a foundation for future hypothesis formation:
“Despite these limitations, this exploratory study is useful for providing a foundation to inform further hypothesis formation about caries and malnutrition in Nepal. Future research should use more robust sampling and statistical
methodology to better understand the directionality of sugar consumption, caries and nutrition-related disease among
Nepali children.” (Lines 280-284)
4. The age distribution of subject in this study is quiet large. The dental status of preschool and school children are total different, which is temporary and ermanent teeth separately. And the possible causations related to the temporary
and permanent teeth could be different, so the authors should consider two models separately.
Response: In this community-based study, we sampled from children ages 0 to 12 years specifically based on the recommendations of the communities and local NGO (Jevaia Foundation) we worked with. For this exploratorystudy, it was imperative that we made decisions regarding sampling that were community-facing, so that we could better understand the health issues among the majority of children in the communities. Additionally, for analysis, we chose to stratify our age groups according to the two major age strata that are relevant for dental development according to the WHO indicator ages of childhood caries: from 6 months to 6 years of age children have primary teeth, and from 6 to 12 years of age children have mixed dentition of primary and permanent teeth. We justified this methodology in section 2.2. Measures using the WHO reference. and we have also acknowledged the limitation of our stratification in section 4. Discussion.
5. The sample size of the article is so limited. it would be suitable if the authors can increase sample size.
Response: We recognize that ideally we would have a larger sample size but in community-based studies in the rural Global South like ours, it is challenging to acquire a large sample size that we see in well-funded studies. We worked
with a community that is difficult to access both geographically and culturally underscoring the importance of this exploratory study to inform future hypothesis formation about caries and malnutrition among Nepali children. Additionally, for our analysis we used Fisher’s exact test to assess the association between dietary frequency and severe caries – this test is used specifically for small sample sizes (see Section 2.3 Data Analysis). We also discussed the limitations of the small sample size in Section 4. Discussion.
Thank you again very much for your review and reconsideration of our revised manuscript. We look forward to hearing your decision.
Round 2
Reviewer 3 Report
Thank you for your responses. The paper is much complete now.